# Does financial asset allocation term structure affect audit fees? Evidence from China

**Chuan Zhang**[1,2,3]**, Hongdi Nie**[1]*

1 School of Economics and Management, Shanghai Maritime University, Shanghai, China, 2 Shanghai University of Electric Power, Shanghai, China, 3 Shanghai Dianji University, Shanghai, China

* nhd18301760763@163.com

**Data Availability Statement:** The data underlying the results presented in the study are available from CSMAR database. URL:https://data.csmar.com/.

## Abstract

The financialization of real enterprises presents a dilemma for China's economic development. This study examines the impact of the financial asset allocation term structure on audit fees using a sample of Chinese A-share listed companies from 2009 to 2019. It also investigates the mediating role of financial risk and the moderating role of independent director characteristics. The results indicate that higher long-term financial assets is associated with higher audit fees, while short-term financial assets show no significant relationship with audit fees. These findings remain robust after several tests. Financial risk mediates the relationship between long-term financial assets and audit fees. Furthermore, among the characteristics of independent directors, the proportion of female independent directors and those with a financial background negatively moderate the relationship between long-term financial assets and audit fees, while independent directors with an overseas background and academic credentials positively moderate this relationship. Additional analysis reveals that firm size and financing constraints exhibit heterogeneity in their effects. This study contributes to the literature by enhancing our understanding of the factors influencing firms' financial asset allocation and audit fees, and by expanding the literature on the financial risk and characteristics of independent directors.

## 1. Introduction

Chinese real enterprises face a dilemma characterized by shrinking markets, intensifying competition, overcapacity, profit compression, and declining returns on investment [1, 2]. In the context of risk-oriented auditing, scholars have increasingly explored the relationship between financial asset allocation and audit fees. Audit fees primarily depend on two factors: cost inputs and the risk premium associated with the audit process [3]. If the auditor intensifies the assessment of the enterprise's risk, it will increase both the content and complexity of the audit work, thereby raising the audit fees [4, 5]. Unlike core business operations, the process of financial asset allocation in enterprises often involves significant agency conflicts [6], which can turn financial asset allocation into a tool for management to enhance performance and engage in earnings management. This, in turn, exacerbates both operational and financial risks [7]. Consequently, auditing financial asset allocation increases the complexity of

**Funding:** The author(s) received no specific funding for this work.

**Competing interests:** The authors have declared that no competing interests exist.

corporate audits and requires experienced auditors. Particularly, the assessment of enterprises asset risk significantly affects the audit fees [8].

Current studies on the motives behind financial asset allocation primarily focus on the risk-averse motive to safeguard assets and the profit-seeking motive to maximize returns [9, 10]. Financial assets can be classified into short-term and long-term categories based on their term structure, which results the financialization of enterprises show different characteristics of the term structure [11]. According to capital demand theory, short-term financial assets, as substitutes for cash, possess high liquidity and serve as a buffer for enterprises to manage financial crises, thereby mitigating corporate risk [12], not a primary factor in increasing corporate audit fees. Long-term financial assets are primarily long-term investments with high returns, aligning with the profit-seeking motive behind financial asset allocation [11]. High returns entail high risks, and long-term financial asset allocation exposes enterprises to greater risks due to longer time horizons, more complex business structures, and a higher degree of uncertainty in both internal and external environments [4, 8]. When an enterprise faces the need to quickly replenish funds for unexpected situations, long-term financial assets lack sufficient liquidity, making it difficult for the enterprise to convert these assets into cash in a timely manner, particularly when confronted with financing constraints and investment opportunities. Moreover, excessive long-term financial asset allocations increase the complexity of the audit process, thereby exacerbating the asset risk faced by the enterprise, which in turn raises audit costs [8]. Therefore, the primary reason financial asset allocation affects audit fees is the associated risk. This study aims to investigate whether short-term or long-term financial asset allocation poses greater risks to enterprises, focusing on the term structure of financial asset allocation and analyzing the underlying mechanisms.

Financial risk is a significant factor influencing enterprise audit fees [8]. To produce an objective and accurate audit report, auditors must conduct extensive work to identify the financial risks faced by the enterprise [7]. Simultaneously, managers may attempt to conceal these risks, further complicating the audit process. When an enterprise faces higher financial risks, audit fees are likely to increase [13]. The primary objective of an enterprise's financial asset allocation is to enhance earnings, which inherently involves financial risk. In the term structure of financial asset allocation, long-term financial assets are associated with high returns and high risks [11], suggesting that the financial risk primarily stems from long-term asset allocation. Therefore, financial risk influences the relationship between the term structure of financial assets and audit fees. In recent years, Chinese government and investors have increasingly emphasized the monitoring and decision-making roles of independent directors. The governance of independent directors influences the relationship between the financial asset allocation term structure and audit fees. However, the direction and magnitude of this influence depend on the characteristics of the independent directors. Consequently, different characteristics of independent directors may have varying impacts on the relationship between the financial asset allocation term structure and corporate audit fees [14].

Based on the above analysis, this study examines the relationship between the financial asset allocation term structure and audit fees, using a sample of Chinese A-share listed companies from 2009 to 2019. It also explores the mediating role of financial risk and the moderating role of independent directors' characteristics. The findings reveal that short-term financial asset allocation has an insignificant impact on audit fees, whereas long-term financial asset allocation significantly increases audit fees, serving as the primary driver of rising corporate audit costs. This effect is more pronounced in large firms and those with high financing constraints. Additionally, financial risk is found to partially mediate the effect of long-term financial asset allocation on audit fees. The proportion of female independent directors and financial expertise negatively moderate the relationship between long-term financial assets and

audit fees, while overseas experience and academic background positively moderate this relationship.

This study makes several potential contributions. Firstly, it examines the relationship between financial asset allocation and audit fees from two dimensions: short-term and long-term financial assets, thereby extending the existing research on the relationship between financial asset allocation and audit fees in Chinese firms. Secondly, from the perspective of risk-oriented audit fees, this study finds that within the financial asset allocation term structure, long-term financial asset allocation exposes firms to higher financial risks, which is the primary driver of increased audit fees. This finding provides a rational explanation for the differences in audit fee levels among Chinese listed firms from the perspective of risk-oriented, enriching the literature on the factors influencing audit fee variations. Thirdly, it expands the research on independent director characteristics by examining their role in decision-making and monitoring during firms' financial asset allocation. It also offers empirical insights into preventing the risks associated with financial asset allocation in Chinese firms. This study provides a theoretical foundation for understanding the relationship between the financial asset allocation term structure and audit fees in Chinese firms. Additionally, it offers empirical evidence on strategies corporate managers can adopt to mitigate financial asset allocation risks and reduce audit fees. The findings also have practical implications for accounting firms and government agencies.

## 2. Theoretical analysis and research hypotheses

Corporate financial asset allocation is primarily driven by two motives: the preventive saving motive and the profit-seeking motive [15]. Preventive saving motive is mainly manifested in alleviating financing constraints by allocating more liquid short-term financial assets to cope with the enterprise's capital demand in unexpected situations, which will reduce the enterprise's asset risk. In contrast, the profit-seeking motive involves allocating less liquid, higher-yield financial assets to achieve excess returns, thereby increasing the enterprise's asset risk [16]. Based on the risk-oriented perspective of audit fees, the primary driver of increased audit fees is the level of long-term financial asset allocation [13]. This raises the question: does the coexistence of precautionary and profit-seeking motives arise from differences in the maturity structure of financial asset allocation? To address this, the study examines the impact of firms' financial asset allocation on audit fees by categorizing financial assets into two dimensions: short-term and long-term.

Financial assets, particularly short-term financial assets, are characterized by high liquidity, which enhances enterprises' ability to manage risk [17]. To address the high costs and financing challenges associated with cash flow risks and external financing constraints [18], enterprises often hold short-term financial assets as a precautionary measure. When a company's core operations require funds or when it urgently needs liquidity to repay maturing debt, it can quickly liquidate short-term financial assets. This helps reduce financial distress costs [12], replenish the necessary liquidity, strengthen the company's refinancing capacity, and effectively mitigate both operational and financial risks [19]. The allocation of short-term financial assets by enterprises contributes to risk prevention. Consequently, the audited firm holds a greater advantage in negotiations with the accounting firm, which may lead to a reduction or stabilization of audit fees. Based on above analysis, the paper proposes Hypothesis 1:

**H1**: Allocation of short-term financial asset reduces audit fees.

Unlike short-term financial asset allocations, which are primarily risk-averse, the profit motive may drive firms to allocate resources to long-term financial assets with higher yields [2,

11]. However, long-term financial assets lack liquidity and the ability to be quickly liquidated. Higher returns are associated with higher risks, and profit-seeking motives often lead firms to allocate a larger portion of their assets to high-risk, long-term financial investments. When faced with external financing constraints, an enterprise's financial risk increases [20]. Additionally, a large allocation of long-term financial assets in high-risk sectors such as finance and real estate further exacerbates financial risk. The presence of these assets also creates a "risk contagion effect," leading to higher audit fees [8]. According to the insurance hypothesis, investors view auditors as guarantors of investment assets and as responsible for covering investment losses [21, 22]. The rational economic man hypothesis posits that auditors have a strong incentive to avoid risks and make corresponding risk control decisions [23]. Compared to short-term financial assets, long-term financial assets are more complex, involving larger foreign investments in both amount and scope [24]. Consequently, auditing firms must allocate more human and material resources to audit these assets. The sustained high-quality development of real enterprises relies on equipment upgrades, reconstruction, and the research and development of new products and technologies. However, if enterprises excessively hold long-term financial assets with low liquidity and high conversion costs, these capital needs may not be effectively met [25]. This can impact the future performance of the entity's core business, leading to financial unsustainability [26, 27], and result in insufficient collateral for obtaining external credit or offsetting maturing debts, which in turn increases both business and financial risks [28]. To mitigate potential future reputation, litigation, and compensation losses [29], auditors are incentivized to adopt a more cautious professional approach and make more conservative decisions for entities holding a larger proportion of long-term financial assets. This may include charging higher audit fees to compensate for costs and to account for risk premiums [30]. Based on this analysis, the paper proposes the hypothesis 2:

**H2**: Allocation of long-term financial assets increases audit fees.

## 3. Research design

### 3.1 Sample selection and data source

This study uses Chinese A-share listed companies as the research sample, with a time frame spanning from 2009 to 2019. The selected period aims to minimize the effects of the 2008 financial crisis and the 2020 COVID-19 pandemic. The initial sample was processed according to the following criteria: (1) exclusion of ST and PT listed companies; (2) exclusion of companies in the financial, insurance, and real estate sectors; (3) exclusion of companies with missing key variables; (4) exclusion of outliers, such as bankruptcies. After applying these filters, the final sample includes 8,547 annual observations. In addition, in order to reduce the impact of data extremes on the research results, continuous variables are reduced-tailed at the 1% and 99% percentile. Independent directors characteristics were also sourced from CSMAR and manually cleaned and screened to create the final sample.

### 3.2 Definition of variables

**Dependent variable.** Audit Fees [31], which is measured by the natural logarithmic computation of the audit fee of listed companies for the year, denoted as LNFEE.

**Independent variables.** Financial Asset Allocation Term Structure. Drawing on previous studies [25, 32], this study includes trading financial assets, available-for-sale financial assets, held-to-maturity investments, loans and advances granted, derivative financial instruments, long-term equity investments, and investment real estate in the scope of financial asset allocation. According to the liquidity level, this study distinguishes between financial

assets with different maturities, short-term financial assets (SFin) = trading financial assets/total assets and long-term financial assets (CFin) = (available-for-sale financial assets + held-to-maturity investments + loans and advances issued + derivative financial instruments + long-term equity investments + investment real estate)/total assets.

**Mediating variable.** Financial Risk, this study employs Z-score to measure financial risk [33].

**Moderating variables.** Independent Directors Characteristics. The moderating variables include the proportion of female independent directors (FID), the proportion of independent directors with financial backgrounds (FBID), the proportion of independent directors with overseas backgrounds (OID), and the presence of academic backgrounds (AEX), as defined in Table 1.

Control variables: with reference to the existing audit fee studies [34], this study selects the control variables. The specific definitions of these control variables are provided in Table 1.

### 3.3 Model design

To investigate the impact of the financial asset allocation term structure on audit fees, Model (1) is developed to test Hypotheses H1 and H2:

$$LNFEE_{i,t} = \beta_0 + \beta_1 Fin_{i,t} + \beta_2 CONTROLS_{i,t} + Year + Industry + \varepsilon_{i,t} \tag{1}$$

**Table 1. Definition of variables.**

| Variable type | Variable name | Variable Definition |
|---|---|---|
| Dependent variable | Audit fees (LNFEE) | Natural logarithm of current year audit fees |
| Independent variable | CFin | Long-term financial assets |
| | SFin | Short-term financial assets |
| Mediating variable | Financial risk | Z-score |
| Moderating variable | FID | The proportion of female |
| | FBID | The proportion of financial backgrounds |
| | OID | The proportion of overseas backgrounds |
| | AEX | The proportion of academic backgrounds |
| Control variables | Audit opinion (Opin) | Dummy variable, 1 if the audit opinion is non-standard, 0 otherwise |
| | Big10 | Audit organization for the CICPA issued the "2019 Annual Comprehensive Evaluation of Accounting Firms Top 100 Information" top ten take 1, otherwise 0 |
| | Change | Dummy variable, 1 for change of auditor during the period, 0 otherwise |
| | Soe | Dummy variable, SOEs take 1, otherwise 0 |
| | Size | Natural logarithm of total assets at the end of the period |
| | Lev | Ratio of total liabilities at the end of the period to total assets at the end of the period |
| | Roa | Ratio of net profit to average total assets at the end of the period and at the beginning of the period |
| | Reca | Ratio of ending inventory to total assets |
| | Inva | Ratio of accounts receivable to total assets at end of period |
| | First1 | Value of proportion of shares held by the largest shareholder (%) |
| | First2 | Sum of shareholding values of the second to tenth largest shareholders (%) |
| | Ind/Year | Dummy variable |

**Table 2. Descriptive statistics.**

| Variables | N | Mean | Sd | min | p25 | P50 | P75 | max |
|---|---|---|---|---|---|---|---|---|
| LNFEE | 8547 | 13.769 | 0.754 | 12.429 | 13.253 | 13.688 | 14.194 | 16.380 |
| CFin | 8547 | 0.071 | 0.102 | 0.000 | 0.007 | 0.031 | 0.086 | 0.542 |
| SFin | 8547 | 0.004 | 0.017 | 0.000 | 0.000 | 0.000 | 0.000 | 0.121 |
| Opin | 8547 | 0.029 | 0.167 | 0.000 | 0.000 | 0.000 | 0.000 | 1.000 |
| Big10 | 8547 | 0.469 | 0.499 | 0.000 | 0.000 | 0.000 | 1.000 | 1.000 |
| Change | 8547 | 0.093 | 0.290 | 0.000 | 0.000 | 0.000 | 0.000 | 1.000 |
| Soe | 8547 | 0.575 | 0.494 | 0.000 | 0.000 | 1.000 | 1.000 | 1.000 |
| Lev | 8547 | 0.469 | 0.196 | 0.069 | 0.319 | 0.475 | 0.615 | 0.912 |
| Roa | 8547 | 0.041 | 0.057 | -0.175 | 0.013 | 0.035 | 0.067 | 0.221 |
| First1 | 8547 | 34.391 | 14.445 | 9.130 | 22.944 | 32.515 | 44.931 | 73.132 |
| First2 | 8547 | 19.747 | 12.127 | 1.929 | 9.784 | 18.089 | 27.873 | 52.057 |
| Reca | 8547 | 0.104 | 0.095 | 0.000 | 0.028 | 0.079 | 0.152 | 0.437 |
| Inva | 8547 | 0.139 | 0.112 | 0.000 | 0.059 | 0.116 | 0.187 | 0.554 |
| Size | 8547 | 22.304 | 1.284 | 19.371 | 21.407 | 22.183 | 23.081 | 26.094 |

The independent variable Fin, includes CFin and SFin, representing the financial asset allocation term structure of real firms.

## 4. Empirical results and analysis

### 4.1 Descriptive statistics

Table 2 presents the descriptive statistics of the main variables. The maximum value of long-term financial assets is 0.542, while the maximum value of short-term financial assets is 0.121, with a minimum value of 0.000 for both. These results highlight significant variations in financial asset allocation among different entities, providing a robust foundation for this study. The mean value of audit opinion is 0.029, indicating that approximately 2.9% of firms received a non-standard audit opinion. The mean value of auditor type is 0.469, suggesting that about 46.9% of firms engaged auditors ranked in the top ten in composite rankings. Additionally, the mean value of shareholding nature is 0.575, indicating that 57.5% of the firms are state-owned enterprises. The statistical values of other variables align closely with findings from previous studies.

### 4.2 Baseline regression results

Table 3 presents the empirical results on the relationship between financial asset allocation term structure and audit fees. Columns (1)–(2) show that the coefficients for short-term financial assets are insignificant in both univariate regression and regression that include control variables and account for industry and year effects, indicating that H1 is not supported. This suggests that the holding of short-term financial assets does not significantly influence audit fees. The strong liquidity of short-term financial assets allows firms to flexibly utilize them to address various business needs, enhancing their resilience to risk.

As shown in columns (3)–(4), the coefficients for long-term financial assets are significantly positive at the 1% level. This finding supports H2, and highlights the profit-seeking motives of enterprises. The lower liquidity and exposure to capital market fluctuations associated with long-term financial assets increase both business and financial risks, leading to higher audit fees. The main regression results further confirm that long-term financial assets exert a stronger positive effect on audit fees due to heightened firm risk driven by profit-seeking

**Table 3. Empirical results.**

| Variables | (1) LNFEE | (2) LNFEE | (3) LNFEE | (4) LNFEE |
|---|---|---|---|---|
| SFin | 0.758 | 0.250 | | |
| | (1.54) | (0.84) | | |
| CFin | | | 0.408*** | 0.449*** |
| | | | (5.13) | (8.96) |
| Opin | | 0.205*** | | 0.200*** |
| | | (7.06) | | (6.91) |
| Big10 | | 0.191*** | | 0.193*** |
| | | (19.86) | | (20.12) |
| Change | | -0.0627*** | | -0.0601*** |
| | | (-3.91) | | (-3.76) |
| Soe | | -0.0266** | | -0.0277** |
| | | (-2.44) | | (-2.55) |
| Lev | | 0.0533* | | 0.0848*** |
| | | (1.66) | | (2.65) |
| Roa | | -0.405*** | | -0.403*** |
| | | (-4.22) | | (-4.22) |
| First1 | | 0.000778** | | 0.00127*** |
| | | (1.97) | | (3.19) |
| First2 | | 0.00357*** | | 0.00428*** |
| | | (7.93) | | (9.42) |
| Reca | | 0.0339 | | 0.113* |
| | | (0.55) | | (1.83) |
| Inva | | -0.0137 | | 0.0459 |
| | | (-0.27) | | (0.91) |
| Size | | 0.423*** | | 0.420*** |
| | | (84.82) | | (84.57) |
| Cons | 13.77*** | 4.088*** | 13.74*** | 4.065*** |
| | (45.17) | (33.63) | (86.37) | (33.61) |
| Ind/Year | no | yes | no | yes |
| N | 8547 | 8547 | 8547 | 8547 |
| $R^2$ | 0.000 | 0.687 | 0.003 | 0.690 |

Note: t-statistics in parentheses

*, **, *** denote statistically significant at 10%, 5%, and 1%.

motivations. In contrast, no significant relationship is observed between short-term financial assets and audit fees. Therefore, this study focuses on examining the mediating effect of financial risk and the moderating effect of independent directors characteristics on the relationship between long-term financial assets and audit fees.

## 4.3 Robustness testing

**4.3.1 Replacement of measures of the independent variable.** Drawing on previous research [26], this study excludes loans and advances issued from the calculation of long-term financial assets. The results, presented in Table 4, columns (1)–(2) indicate that the coefficients for SFin1 are insignificant, while the coefficients for CFin1 remain significantly positive at the

**Table 4. Robustness test.**

| Variables | (1) LNFEE | (2) LNFEE | (3) LNFEE | (4) LNFEE | (5) LNFEE | (6) LNFEE | (7) LNFEE | (8) LNFEE |
|---|---|---|---|---|---|---|---|---|
| SFin1 | 0.250 | | | | | | | |
| | (0.84) | | | | | | | |
| CFin1 | | 0.458*** | | | | | | |
| | | (9.06) | | | | | | |
| L1.SFin | | | 0.203 | | | | | |
| | | | (1.02) | | | | | |
| L1.CFin | | | | 0.319*** | | | | |
| | | | | (3.47) | | | | |
| L2.SFin | | | | | 0.143 | | | |
| | | | | | (0.28) | | | |
| L2.CFin | | | | | | 0.204** | | |
| | | | | | | (2.01) | | |
| SFin | | | | | | | 0.172 | |
| | | | | | | | (0.69) | |
| CFin | | | | | | | | 0.134*** |
| | | | | | | | | (2.99) |
| Cons | 4.088*** | 4.064*** | 3.875*** | 3.927*** | 3.106*** | 3.437*** | 2.517*** | 3.313*** |
| | (33.63) | (33.61) | (28.36) | (27.54) | (22.61) | (24.58) | (18.54) | (23.16) |
| Controls | yes | yes | yes | yes | yes | yes | yes | yes |
| Ind/Year | yes | yes | yes | yes | yes | yes | yes | yes |
| N | 8547 | 8547 | 8547 | 8547 | 8547 | 8547 | 7942 | 7942 |
| $R^2$ | 0.687 | 0.690 | 0.514 | 0.506 | 0.462 | 0.493 | 0.416 | 0.484 |

Note: t-statistics in parentheses

*, **, *** denote statistically significant at 10%, 5%, and 1%.

1% level. These findings are consistent with the main regression results, reinforcing the robustness of the analysis.

**4.3.2 Lagged independent variable.** Lagged the independent variable by one and two periods, the results are shown in Table 4. Columns (3)-(6) show the results that the coefficients of L1.SFin and L2.SFin are insignificant, the coefficients of L1.CFin and L2.CFin are significantly positive at the 1% level and 5% level. These findings confirm the robustness of the main regression results.

**4.3.3 Adjusting the sample interval.** To mitigate the potential impact of the financial crisis, data from 2009 and 2010 were excluded from the analysis. The results, presented in columns (7)-(8) of Table 4, show that the coefficients for CFin remain significantly positive at the 1% level. These findings further confirm the robustness of the main regression results.

**4.3.4 Endogeneity test.** Instrumental Variables (IV) Estimation. Following prior research [1, 26], "investment return/net profit" was selected as the instrumental variable (IV) to address potential endogeneity concerns in the study's findings. Investment return is strongly correlated with the level of financial asset allocation but has no intrinsic relationship with auditor input or effort. This ensures that the IV is uncorrelated with audit fees, satisfying the basic conditions for a valid instrumental variable. At the same time, Hansen J statistic P value is 0.167, F value is 549.457, LM statistic is 206.592, confirm the validity of the instrumental variable. Columns (1) and (2) in Table 5 present the results of the IV method. In the first-stage regression, the coefficient of the IV in column (1) is 0.0219, which is significantly positive at the 1% level,

**Table 5. Endogeneity test.**

| Variables | (1) | (2) | (3) | (4) | (5) | (6) | (7) | (8) |
|---|---|---|---|---|---|---|---|---|
| | IV-2SLS | | GMM | GMM | | | | |
| | CFin | LNFEE | LNFEE | LNFEE | LNFEE | LNFEE | LNFEE | LNFEE |
| IV | 0.0219*** | | | | | | | |
| | (12.78) | | | | | | | |
| CFin | | 0.301*** | | | | | | |
| | | (5.13) | | | | | | |
| SFin | | | 0.109 | | 0.312 | | 0.407 | |
| | | | (0.96) | | (1.16) | | (1.27) | |
| CFin | | | | 0.016** | | 0.330** | | 0.493** |
| | | | | (1.99) | | (1.97) | | (2.41) |
| Cons | 0.0571** | 4.014*** | 1.976** | 2.013*** | 1.476*** | 3.540*** | 2.219*** | 3.131*** |
| | (2.20) | (27.70) | (2.03) | (16.03) | (23.54) | (26.18) | (18.33) | (19.16) |
| Controls | yes | yes | yes | yes | yes | yes | yes | yes |
| Ind/Year | yes | yes | yes | yes | yes | yes | yes | yes |
| Company | no | no | no | no | yes | yes | no | no |
| Ind -Year | no | no | no | no | no | no | yes | yes |
| N | 8547 | 8547 | 8547 | 8547 | 8547 | 8547 | 8547 | 8547 |
| $R^2$ | 0.246 | 0.680 | 0.427 | 0.549 | 0.436 | 0.548 | 0.477 | 0.501 |

Note: t-statistics in parentheses

*, **, *** denote statistically significant at 10%, 5%, and 1%.

indicating a strong positive relationship between the IV and the endogenous variable. In the second-stage regression, the coefficient of long-term financial assets in column (2) is 0.301, also significantly positive at the 1% level, demonstrating that audit pricing increases substantially with higher levels of long-term financial assets. These results are consistent with the main regression findings.

Generalized Method of Moments (GMM). GMM was used to re-evaluate the results. Following prior research, the lagged one-period independent variable was selected as the instrumental variable [35]. The results satisfy the requirements for overidentification and autocorrelation tests. As shown in Table 5, column (3) reports a regression coefficient of 0.109 for short-term financial assets, which is not statistically significant. Column (4) indicates that the regression coefficient for long-term financial assets is significantly positive at the 5% level, further confirming the reliability of the previous findings

Fixed Effects Model. Endogeneity can arise from omitted variables. To address this, the study incorporates year fixed effects, industry fixed effects, and firm fixed effects in the regression model. The results are presented in Table 5. In Column (5), the coefficient for short-term financial assets is insignificant, while in Column (6), the coefficient is significantly positive at the 5% level. Additionally, a high-dimensional fixed effects regression at the "industry-year" level is conducted, as shown in Columns (7) and (8) of Table 5. The results indicate that higher levels of long-term financial assets significantly increase audit fees, confirming the robustness of the main regression findings.

## 4.4 Mechanism analysis

**4.4.1 Financial risks.** The main regression results demonstrate a significant positive relationship between long-term financial assets and audit fees. As analyzed earlier, firms'

**Table 6. Mechanism analysis of financial risks.**

| Variables | (1) LNFEE | (2) Z-score | (3) LNFEE |
|---|---|---|---|
| CFin | 0.607*** | -1.240*** | 0.560*** |
| | (8.72) | (-13.60) | (7.93) |
| Z-score | | | -0.038*** |
| | | | (-3.80) |
| Cons | 7.879*** | 2.917*** | 7.990*** |
| | (49.68) | (14.05) | (49.60) |
| Controls | yes | yes | yes |
| Ind/Year | yes | yes | yes |
| N | 5918 | 5918 | 5918 |
| $R^2$ | 0.645 | 0.638 | 0.646 |

Note: t-statistics in parentheses

*, **, *** denote statistically significant at 10%, 5%, and 1%.

allocation of long-term financial assets increases financial risk, heightens the likelihood of potential audit failures, and leads to higher audit fees. To examine whether the allocation of long-term financial assets affects audit fees by influencing financial risk, this study employs a three-step method and uses the Z-score as a measure of financial risk [33], which is computed as Z-score = (0.717 × working capital + 0.847 × retained earnings + 3.107 × EBIT + 0.42 × total stock market value + 0.998 × sales revenue)/total assets. A higher Z-score indicates lower financial risk. Models (2), (3), and (4) are constructed to further investigate the mediating role of financial risk in the relationship between long-term financial asset allocation and audit fees. The findings confirm that financial risk mediates the effect of long-term financial asset allocation on audit fees.

$$LNFEE_{i,t} = \beta_0 + \beta_1 CFin_{i,t} + CONTROLS_{i,t} + Year + Industry + \varepsilon_{i,t} \tag{2}$$

$$Zscore_{i,t} = \partial_0 + \partial_1 CFin_{i,t} + CONTROLS_{i,t} + Year + Industry + \varepsilon_{i,t} \tag{3}$$

$$LNFEE_{i,t} = \alpha_0 + \alpha_1 CFin_{i,t} + \alpha_2 Zscore_{i,t} + CONTROLS_{i,t} + Year + Industry + \varepsilon_{i,t} \tag{4}$$

Columns (1)–(3) of Table 6 present the regression results for financial risk. In column (2), the coefficient between long-term financial assets and the Z-score is negative and statistically significant at the 1% level, indicating that firms with higher long-term financial assets face greater financial risk. Column (3) investigates the relationship among long-term financial assets, financial risk, and audit fees. The results show that the regression coefficient for long-term financial assets remains significantly positive even after including the financial risk variable. Additionally, the coefficient for the Z-score is significantly negative at the 1% level, confirming that financial risk mediates the relationship between long-term financial assets and audit fees. These findings indicate that higher financial risk leads to increased audit fees.

**4.4.2 Independent directors characteristics.** Independent directors play a critical role in strategy formulation and addressing procedural uncertainties [36–38]. Effective governance by independent directors enhances corporate decision-making efficiency, improves asset allocation decisions, and optimizes the overall corporate asset structure [39]. The characteristics of independent directors define the structure and distribution of their authority, ensuring professional and independent decision-making and monitoring within the

framework of formal governance while safeguarding the interests of all shareholders [36]. Specifically, their governance mechanism enhances both decision-making and oversight capacities. According to higher-order theory, the characteristics of top managers, such as age, gender, educational background, and career experiences, which significantly influence a firm's strategic choices [40]. This study aims to investigate how the individual characteristics of independent directors moderate the relationship between long-term financial assets and audit fees by improving decision-making efficiency. This study identifies gender, financial experience, overseas experience, and academic background as key proxies for the individual characteristics of independent directors. Using Model (5), it examines the moderating effects of the proportion of women (FID), the proportion of directors with financial backgrounds (FBID), the proportion with overseas experience (OID), and the presence of academic backgrounds (AEX) on the positive association between long-term financial assets and audit fees.

In order to test the moderating effect of independent directors characteristics on the relationship between long-term financial assets and audit fees, this study constructs model (5). The moderator variable IDC includes FID, FBID, OID, and AEX.

$$LNFEE_{i,t} = \theta_0 + \theta_1 CFin_{i,t} + \theta_2 IDC_{i,t} + \theta_3 CFin_{i,t}*IDC_{i,t} + \beta_4 CONTROLS_{i,t} + Year + Industry + \varepsilon_{i,t} \quad (5)$$

Table 7 presents the regression results examining the moderating effect of independent directors' characteristics on the relationship between long-term financial assets and audit fees. The coefficients of the interaction terms in columns (1) and (2) are significantly negative at the 1% level, indicating that the proportion of female independent directors and those with financial backgrounds mitigates the positive relationship between long-term financial assets and audit fees. This suggests that independent directors with financial expertise and female independent directors may enhance information transparency, reduce corporate agency costs, and avoid high-risk investments, thereby improving disclosure quality and decreasing management's incentives for financial fraud or violations [41]. The coefficients in columns (3) and (4) are significantly positive at the 1% level, indicating that the proportion of independent directors with overseas backgrounds and the presence of independent directors with academic backgrounds amplify the positive relationship between long-term financial assets and audit fees. This suggests that independent directors with overseas and academic backgrounds are more confident and outspoken in their decision-making.

## 5. Further research

### 5.1 Heterogeneity of firm size

Firm size is a fundamental characteristic influencing firms' investment and financing behaviors. Research indicates that financialization trends are more pronounced in larger enterprises [42]. Large firms typically engage in more complex business activities, possess intricate governance structures, incur higher agency costs, and are more susceptible to risks, all of which can influence auditor behavior. To analyze the impact of long-term financial asset holdings on audit pricing across firms of different sizes, the sample is divided into large and small enterprises based on the median firm size. The regression results, presented in columns (1) and (2) of Table 8, indicate that long-term financial assets increase audit pricing for both large and small firms. However, this effect is more pronounced in larger firms and is statistically significant. The coefficient differences between groups are significant, P-value is 0.007, as confirmed by Fisher's Permutation Test. This suggests that larger firms, with their more complex economic operations, are subject to greater audit adjustments, requiring auditors to gather more evidence, thereby increasing audit fees [43].

**Table 7. Mechanism analysis of independent directors characteristics.**

| Variables | (1) LNFEE | (2) LNFEE | (3) LNFEE | (4) LNFEE |
|---|---|---|---|---|
| CFin | 0.178** | 0.209*** | 0.166*** | 0.153*** |
| | (1.99) | (3.52) | (2.82) | (3.16) |
| FID | -0.324*** | | | |
| | (-2.96) | | | |
| FID*CFin | -0.141*** | | | |
| | (-5.41) | | | |
| FBID | | -0.779*** | | |
| | | (-4.13) | | |
| FBID*CFin | | -0.603*** | | |
| | | (-3.11) | | |
| OBID | | | 0.682*** | |
| | | | (3.28) | |
| OBID*CFin | | | 0.647*** | |
| | | | (2.59) | |
| AEX | | | | 0.657*** |
| | | | | (4.16) |
| AEX*CFin | | | | 0.606*** |
| | | | | (5.17) |
| Cons | 4.051*** | 4.060*** | 4.167*** | 4.112*** |
| | (33.43) | (33.55) | (34.40) | (33.87) |
| Controls | yes | yes | yes | yes |
| Ind/Year | yes | yes | yes | yes |
| N | 8547 | 8547 | 8547 | 8547 |
| $R^2$ | 0.691 | 0.691 | 0.693 | 0.691 |

Note: t-statistics in parentheses

*, **, *** denote statistically significant at 10%, 5%, and 1%.

**Table 8. Heterogeneity analysis.**

| Variables | (1) LNFEE Large-Size | (2) LNFEE Small-Size | (3) LNFEE High-SA | (4) LNFEE Low-SA |
|---|---|---|---|---|
| CFin | 0.452*** | 0.307* | 0.169*** | 0.073 |
| | (3.27) | (1.76) | (4.43) | (1.49) |
| Cons | 5.302*** | 6.096*** | 8.035*** | 2.309*** |
| | (39.643) | (36.08) | (60.08) | (17.47) |
| Controls | yes | yes | yes | yes |
| Ind/Year | yes | yes | yes | yes |
| N | 8547 | 8547 | 8547 | 8547 |
| $R^2$ | 0.515 | 0.454 | 0.438 | 0.541 |
| P | 0.007*** | | 0.003*** | |

Note: t-statistics in parentheses

*, **, *** denote statistically significant at 10%, 5%, and 1%.

## 5.2 Heterogeneity of financing constraints

Firms' financial asset allocations require funding, particularly for long-term financial assets, which demand more substantial financial resources. Consequently, firms make allocation decisions based on the extent of their financing constraints. Firms with greater access to funds and lower financing constraints are more likely to invest in long-term financial assets, whereas firms facing higher financing constraints tend to reduce their holdings of such assets. When firms experience greater financing constraints, management often adopts more aggressive financing strategies, leading to increased firm risk and, consequently, higher audit fees. The sample is categorized into high and low financing constraints based on the SA index. Columns (3) and (4) in Table 8 reveal a more significant positive relationship between long-term financial asset allocation and audit pricing in the high financing constraint group. The coefficient differences between groups are significant, P-value is 0.003, as confirmed by Fisher's Permutation Test. This finding suggests that firms facing higher financing constraints typically adopt aggressive strategies to alleviate financing difficulties, thereby increasing firm risk and audit costs.

## 6. Conclusions

Using a sample of Chinese A-share listed companies from 2009 to 2019, this study examines the impact of the financial asset allocation term structure on audit fees. It also explores the mediating role of financial risk and the moderating effect of independent directors characteristics on the relationship between long-term financial asset allocation and audit fees. The findings reveal that long-term financial assets are significantly positively associated with audit fees, whereas short-term financial assets show no significant relationship. Financial risk is found to play a significant mediating role in the relationship between long-term financial assets and audit fees. Independent directors with overseas or academic backgrounds strengthen the positive relationship between long-term financial assets and audit fees, while the proportion of female directors and directors with financial backgrounds weakens this relationship. Further analysis demonstrates heterogeneity in this relationship: the positive association is more pronounced in larger firms and those with higher financing constraints. The study identifies asset risk as the primary factor linking financial asset allocation to audit fees. Specifically, long-term financial asset allocation significantly contributes to audit fees, whereas short-term financial asset allocation does not. These findings underscore the importance of examining the term structure of financial asset allocation in understanding its impact on audit fees.

China's economy has become an increasingly significant component of global economic development, with Chinese enterprises playing a vital role in driving economic growth. Against the backdrop of the financialization of Chinese real enterprises, this study holds important theoretical and practical implications. First, it extends the discussion on the economic impact of the financial asset allocation term structure of Chinese firms and the determinants of audit costs. Additionally, it uncovers the intrinsic mechanism linking the financial asset allocation term structure to audit fees, emphasizing the need for firms to focus on optimizing their financial asset allocation—particularly long-term financial assets—to mitigate financial risk and reduce audit fees. Furthermore, the study provides a theoretical foundation for strengthening the role of independent directors in Chinese enterprises. Firms should leverage the oversight and decision-making capabilities of independent directors to mitigate risks associated with financial asset allocation, thereby reducing audit supervision and related costs. Accounting firms are encouraged to prioritize understanding the financial asset allocation term structure to determine audit fees more effectively. Lastly, government authorities should

guide real enterprises in optimizing their financial asset allocation structures to foster a virtuous cycle between finance and the real economy.

## Supporting information

**S1 File. Data for variables of this paper.**
(XLSX)

## Acknowledgments

We would like to thank Lixia Liu, Tao Feng, Wenbo Luo, Bo Zhang for their constructive comments and advise on this paper.

## Author Contributions

**Conceptualization:** Chuan Zhang, Hongdi Nie.

**Data curation:** Chuan Zhang, Hongdi Nie.

**Formal analysis:** Chuan Zhang, Hongdi Nie.

**Funding acquisition:** Hongdi Nie.

**Investigation:** Chuan Zhang.

**Methodology:** Chuan Zhang.

**Project administration:** Hongdi Nie.

**Resources:** Chuan Zhang.

**Software:** Hongdi Nie.

**Supervision:** Chuan Zhang.

**Validation:** Chuan Zhang.

**Visualization:** Chuan Zhang, Hongdi Nie.

**Writing – original draft:** Hongdi Nie.

**Writing – review & editing:** Chuan Zhang.

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
