## [Decision Letter · Decision Letter 0]

6 Nov 2024

PONE-D-24-38347Financial asset allocation term structure, Independent directors Characteristics and Audit fees: Evidence from ChinaPLOS ONE

Dear Dr. Nie,

Thank you for submitting your manuscript to PLOS ONE. After careful consideration, we feel that it has merit but does not fully meet PLOS ONE’s publication criteria as it currently stands. Therefore, we invite you to submit a revised version of the manuscript that addresses the points raised during the review process. After my own reading of the study, I agree with the reviewer's evaluation. I would like to highlight that the theoretical foundation regarding audit fees requires further development in its current form. There exists a substantial body of literature on the determinants of audit fees that has been well-established over several decades, and I would encourage its incorporation in your revision. In particular, the link between asset risk and audit fees would benefit from a deeper analysis. To strengthen the foundation of your discussion on audit fee determinants, it may be useful to consult comprehensive literature reviews, such as Widmann et al.’s study, What is it going to cost? Empirical evidence from a systematic literature review of audit fee determinants Management Review Quarterly, 2021, https://doi.org/10.1007/s11301-020-00190-w). Overall, I consider the selection of literature to be very limited. More emphasis should be placed on embedding the study in the current state of research.

Thank you for your efforts on this research, and I look forward to seeing how the manuscript develops in response to these recommendations.

We look forward to receiving your revised manuscript.

Kind regards,

Florian Follert

Academic Editor

PLOS ONE

Journal Requirements:

Reviewers' comments:

Reviewer's Responses to Questions

**Comments to the Author**

1. Is the manuscript technically sound, and do the data support the conclusions?

Reviewer #1: Partly

2. Has the statistical analysis been performed appropriately and rigorously? 

Reviewer #1: No

3. Have the authors made all data underlying the findings in their manuscript fully available?

Reviewer #1: No

4. Is the manuscript presented in an intelligible fashion and written in standard English?

Reviewer #1: Yes

5. Review Comments to the Author

Reviewer #1: 1.The authors should enhance the motivation of the study, it is why to study association of financial assets on audit costs, and why utilize the Chinese context to analyze this issue. The paper does not have strong research significance and these issues need to be highlighted in the research contribution.

2.The research findings should be clearly written in the Introduction.

3. The analysis of auditing in theoretical analysis and research hypotheses is insufficient, and the reasons for the differences in the level of audit fees should be analyzed from the perspective of risk-oriented auditing.

4. Financial asset allocation has many dimensions, why did the authors only analyze Short-term and Long-term financial asset allocation?

5. Why the authors chose these moderator variables for Mechanism analysis, further tests are not very relevant to the main logic of the paper's hypothesis, and more tests that are more relevant to the main logic of the paper should be added.

6. First1 and First2 should not be placed in the control variables at the same time and one of them should be deleted.

7.Why choose “Investment return/net profit” as instrumental variables? The choice of instrumental variables does not seem to be very convincing.

8. The endogeneity problem of the article is quite obvious, more endogeneity tests should be added, such as PSM.

9. 5.2 Financial risks mechanism test should be put into 4.4 Mechanism analysis.

10. The literature cited in the paper is not so convincing, and many of them need to be replaced.

6. PLOS authors have the option to publish the peer review history of their article (what does this mean?). If published, this will include your full peer review and any attached files.

Reviewer #1: No

---

## [Author Response · Author response to Decision Letter 0]

3 Dec 2024

Response to Reviewers

Dear Editors and Reviewers:

Thanks a lot for your letter and the reviewers’ comments concerning our manuscript entitled “Financial asset allocation term structure, Independent directors Characteristics and Audit fees: Evidence from China”. We are happy to respond to your comments and suggestions, and grateful to take this as a great opportunity to update our information and try our best to meet the high standards of PLOS ONE. See the document “Response to Reviewers” for more detailed change markup. The main corrections in the paper and the responses to the editor’s and reviewers’ comments, point-by-point, are as follows:

Editor comments:

1.After my own reading of the study, I agree with the reviewer's evaluation. I would like to highlight that the theoretical foundation regarding audit fees requires further development in its current form. There exists a substantial body of literature on the determinants of audit fees that has been well-established over several decades, and I would encourage its incorporation in your revision. In particular, the link between asset risk and audit fees would benefit from a deeper analysis.

R: Thank you so much! See the document “Response to Reviewers” for more detailed change markup.

We believe that your valuable comments are very helpful in improving the quality of the manuscript and have expanded our research vision and research ideas, giving us a deeper knowledge of current research. Based on your comments, we have strengthened the analysis of the link between asset risk and audit fees in the introduction and theoretical research sections. References to more compelling articles on current research have been updated in the body of the manuscript and references.

Firstly, we updated our analysis of the link between asset risk and audit fees in the introduction:

Real enterprises are facing the dilemma of shrinking market, intensifying competition, overcapacity, profit compression, and declining return on investment [1, 2]. In the background of risk-oriented auditing, scholars have gradually begun to explore the relationship between financial asset allocation and audit fees. Audit fees theory holds that audit fee mainly depends on two parts: cost input and risk premium in the audit process [3]. If the auditor enhances the assessment of the risk of the enterprise, it will lead to an increase in the content and complexity of the audit work and raise the audit fee [4, 5]. Distinguished from the general main business, there is a serious agency conflict hidden in the process of financial asset allocation by enterprises [6], which leads to the financial asset allocation becoming a tool for management to glorify the performance and earnings management, and exacerbates the operational and financial risks of the enterprise [7], so in the audit of the financial asset allocation of the enterprise, it requires experienced auditors to carry out the audit. At the same time, it increases the complexity of corporate auditing. Especially for the assessment of enterprise asset risk, it will affect the audit cost of the enterprise [8].

The current study of financial asset allocation motives mainly includes the saving motive to prevent risk, and the profit-seeking motive to obtain more returns [9, 10]. Financial assets can be divided into short-term and long-term according to their term structure, which makes the financialization of enterprises show different characteristics of the term structure [11]. According to the theory of capital demand, short-term financial assets, as a substitute for cash, have strong liquidity and liquidity ability, and the trading financial assets held by enterprises are a buffer to deal with financial crisis, which can prevent corporate risk [12], and are not the main factor to enhance corporate audit fees. Long-term financial assets are mostly long-term investments with high returns, in line with the profit-seeking motive of financial asset allocation [11]. High returns imply high risks, and long-term financial asset allocation will expose enterprises to more risks due to the long cycle, more complex business structure, and high degree of uncertainty in the internal and external environments they face [4, 8]. When the enterprise faces the need to replenish the funds required by the unexpected situation in the short term, the long-term financial assets lack sufficient liquidity, and it is difficult for the enterprise to change the nature of the funds in a timely manner in the face of the financing constraints and investment opportunities, and at the same time, too many long-term financial asset allocations increase the complexity of the auditing business, which in turn exacerbates the level of the risk of the assets faced by the entity enterprise, thus enhancing the audit costs [8].

Secondly, we have added to the introduction a section about the relationship between financial asset allocation and financial risk:

Financial risk is an important factor affecting the audit fee of enterprises[8]. In order to make an objective and fair audit report, the auditor will carry out a lot of audit work to find out the enterprises' financial risks [7]. At the same time, the managers of the enterprise will cover up the financial risks of the enterprise, which increases the difficulty of the audit work. When the enterprise faces higher financial risks, it will increase the audit fees[18]. The main purpose of the enterprise's financial asset allocation is to increase earnings, which will bring financial risk. In the term structure of financial asset allocation, long-term financial assets are characterized by high return and high risk [11], indicating that this effect mainly comes from long-term financial asset allocation, so financial risk will have an impact between the term structure of financial assets and audit costs.

At the same time, We have emphasized in the theoretical analysis and research hypotheses section that asset risk from long-term financial asset allocations is a significant cause of higher audit fees:

There are two main motives for corporate financial asset allocation: the preventive saving motive and the profit-chasing motive [19]. Preventive saving motive is mainly manifested in alleviating financing constraints by allocating more liquid short-term financial assets to cope with the enterprise's capital demand in unexpected situations, which will reduce the enterprise's asset risk; profit-chasing motive is manifested in allocating less liquid but higher-returning financial assets to obtain the excess rate of return [20], which will enhance the enterprise's asset risk. Based on the risk orientation of audit fees [18], the main reason for increasing audit fees should be the level of long-term financial asset allocation in financial asset allocation.

Finally, we refer to recent studies in recent years.

Thanks again and Best regards!

Reviewers' comments:

Reviewer #1: 

1.The authors should enhance the motivation of the study, it is why to study association of financial assets on audit costs, and why utilize the Chinese context to analyze this issue. The paper does not have strong research significance and these issues need to be highlighted in the research contribution.

R: Thank you so much! See the document “Response to Reviewers” for more detailed change markup.

Your comments are very important to the quality of our article, and we are honored to have been able to improve it with your help. We have strengthened the motivation of the study in the introduction and theoretical analysis. Meanwhile, the significance of the study was elaborated in the research contribution, and changes were made in the manuscript as follows:

Firstly, we enhance the motivation of the study in Introduction section:

Chinese real enterprises are facing the dilemma of shrinking market, intensifying competition, overcapacity, profit compression, and declining return on investment [1, 2]. In the background of risk-oriented auditing, scholars have gradually begun to explore the relationship between financial asset allocation and audit fees. Audit fees theory holds that audit fee mainly depends on two parts: cost input and risk premium in the audit process [3]. If the auditor enhances the assessment of the risk of the enterprise, it will lead to an increase in the content and complexity of the audit work and raise the audit fee [4, 5]. Distinguished from the general main business, there is a serious agency conflict hidden in the process of financial asset allocation by enterprises [6], which leads to the financial asset allocation becoming a tool for management to glorify the performance and earnings management, and exacerbates the operational and financial risks of the enterprise [7], so in the audit of the financial asset allocation of the enterprise, it requires experienced auditors to carry out the audit. At the same time, it increases the complexity of corporate auditing. Especially for the assessment of enterprise asset risk, it will affect the audit cost of the enterprise [8].

The current study of financial asset allocation motives mainly includes the saving motive to prevent risk, and the profit-seeking motive to obtain more returns [9, 10]. Financial assets can be divided into short-term and long-term according to their term structure, which makes the financialization of enterprises show different characteristics of the term structure [11]. According to the theory of capital demand, short-term financial assets, as a substitute for cash, have strong liquidity and liquidity ability, and the trading financial assets held by enterprises are a buffer to deal with financial crisis, which can prevent corporate risk [12], and are not the main factor to enhance corporate audit fees. Long-term financial assets are mostly long-term investments with high returns, in line with the profit-seeking motive of financial asset allocation [11]. High returns imply high risks, and long-term financial asset allocation will expose enterprises to more risks due to the long cycle, more complex business structure, and high degree of uncertainty in the internal and external environments they face [4, 8]. When the enterprise faces the need to replenish the funds required by the unexpected situation in the short term, the long-term financial assets lack sufficient liquidity, and it is difficult for the enterprise to change the nature of the funds in a timely manner in the face of the financing constraints and investment opportunities, and at the same time, too many long-term financial asset allocations increase the complexity of the auditing business, which in turn exacerbates the level of the risk of the assets faced by the entity enterprise, thus enhancing the audit costs [8]. Therefore, the main reason why financial asset allocation affects the audit fee is the risk posed. This study tries to investigate whether it is the short-term or long-term financial asset allocation that brings risks to enterprises from the perspective of the term structure of financial asset allocation and analyze the inner mechanism.

Secondly，we explain why utilize the Chinese context to analyze this issue in introduction section.

Chinese real enterprises are facing the dilemma of shrinking market, intensifying competition, overcapacity, profit compression, and declining return on investment [1, 2]. 

Independent directors Characteristics refer to the distribution of independent directors' power shaped by institutional regulations or organizational design, defining the multidimensional situation of the board's structure and status, which are designed to ensure that the independent directors provide the board with a more professional and independent decision-making and oversight capacity within the framework of the formal power, and play a decisive role in strategy formulation and resolving procedural uncertainties [13-15]. Based on modern organizational theory, the effectiveness of board governance affects the structural configuration of corporate assets. Good independent directors governance can improve the efficiency of corporate decision-making [16], enhance the effectiveness of asset allocation decisions, and optimize corporate asset structure allocation. In recent years, there have been frequent cases of financial fraud by listed companies in China. The government and investors have increasingly emphasized the monitoring and decision-making role of independent directors. Thus, independent directors governance may have an impact on the relationship between financial asset allocation term structure and audit fees. However, the direction and extent of this influence are constrained by the independent directors' own characteristics. As a result, different independent directors' characteristics may have different impacts on the relationship between financial asset allocation term structure and corporate audit fees[17]. 

Thirdly, we highlighted research significance in the research contribution.

The possible contributions, firstly, this study explores the relationship between financial asset allocation and audit fees from two dimensions: short-term financial assets and long-term financial assets, expanding the research on the relationship between financial asset allocation and audit fees. Secondly, based on the risk orientation of audit fees, this study finds that in the term structure of financial asset allocation, long-term financial asset allocation leads to the fact that firms will be exposed to higher financial risks, which is the main reason for the increase in firms' audit fees, and enriches the literature on the differences in the level of audit fees under the risk-oriented perspective. Thirdly, it expands the related research on independent director characteristics and explores the role of independent director characteristics in decision-making and monitoring when firms engage in financial asset allocation, which provides empirical experience for preventing the risk of firms' financial asset allocation. 

2. The research findings should be clearly written in the Introduction.

R: Thank you so much!

The research findings have been described in the introduction section and changes have been made in the manuscript as follows:

Based on the above analysis, using a sample of Chinese A-share listed companies in the period 2009-2019 as the research sample, this study examines the relationship between the term structure of financial asset allocation and audit fees, as well as the moderating role of independent directors' characteristics and the mediating role of financial risk. It is found that the relationship between short-term financial asset allocation and audit fees is insignificant, and long-term financial asset allocation significantly contributes to audit fees and is the main reason for the increase in corporate audit fees, and this effect is more significant in large firms and firms with high financing constraints. The proportion of female independent directors and financial background negatively moderated the relationship between long-term financial assets and audit fees, and overseas background and academic background positively moderated the positive relationship between long-term financial assets and audit fees. Financial risk plays a partial mediating role. 

3. The analysis of auditing in theoretical analysis and research hypotheses is insufficient, and the reasons for the differences in the level of audit fees should be analyzed from the perspective of risk-oriented auditing.

R: Thank you so much!

Based on your comments, we have analyzed the reasons for the level of audit fee variance from a risk-oriented perspective and have expressed them in the Introduction section and Theoretical Analysis and Research Hypotheses section as follows:

Firstly, we updated our analysis of the link between asset risk and audit fees in the introduction:

Chinese real enterprises are facing the dilemma of shrinking market, intensifying competition, overcapacity, profit compression, and declining return on investment [1, 2]. In the background of risk-oriented auditing, scholars have gradually begun to explore the relationship between financial asset allocation and audit fees. Audit fees theory holds that audit fee mainly depends on two parts: cost in

---

## [Decision Letter · Decision Letter 1]

22 Dec 2024

PONE-D-24-38347R1Financial asset allocation term structure, Independent directors Characteristics and Audit fees: Evidence from ChinaPLOS ONE

Dear Dr. Nie,

Thank you for submitting your manuscript to PLOS ONE. After careful consideration, we feel that it has merit but does not fully meet PLOS ONE’s publication criteria as it currently stands. Therefore, we invite you to submit a revised version of the manuscript that addresses the points raised during the review process.

We look forward to receiving your revised manuscript.

Kind regards,

Florian Follert

Academic Editor

PLOS ONE

Journal Requirements:

Reviewers' comments:

Reviewer's Responses to Questions

**Comments to the Author**

1. If the authors have adequately addressed your comments raised in a previous round of review and you feel that this manuscript is now acceptable for publication, you may indicate that here to bypass the “Comments to the Author” section, enter your conflict of interest statement in the “Confidential to Editor” section, and submit your "Accept" recommendation.

Reviewer #1: (No Response)

2. Is the manuscript technically sound, and do the data support the conclusions?

Reviewer #1: Partly

3. Has the statistical analysis been performed appropriately and rigorously? 

Reviewer #1: No

4. Have the authors made all data underlying the findings in their manuscript fully available?

Reviewer #1: Yes

5. Is the manuscript presented in an intelligible fashion and written in standard English?

Reviewer #1: No

6. Review Comments to the Author

Reviewer #1: 1. The appropriateness of the term "Audit fees theory" and whether it is clearly defined in the existing literature needs further investigation. It is suggested that the authors provide a more detailed explanation of this theory and review relevant literature.

2. The significance of conducting the study in the context of China should be further emphasized in the contribution section.

3. The current endogeneity tests in the paper appears inadequate. The authors are encouraged to add one or two more endogeneity tests.

4. The title and introduction of the paper place significant emphasis on the influence of independent directors. However, in the mechanism analysis section, the characteristics of independent directors are only considered as one aspect of the analysis. It is suggested that the authors either include the analysis of independent directors' characteristics in the hypothesis development section, or adjust the title and introduction to focus specifically on audit fees and financial asset allocation structures. This would improve the clarity and logical coherence of the paper’s structure.

5. In the "Further Research" section, the authors should include tests for coefficient differences between groups.

6. The overall quality of the writing needs significant improvement.

7. PLOS authors have the option to publish the peer review history of their article (what does this mean?). If published, this will include your full peer review and any attached files.

Reviewer #1: No

---

## [Author Response · Author response to Decision Letter 1]

30 Dec 2024

Response to Reviewers

Dear Editors and Reviewers:

Thanks a lot for your letter and the reviewers’ comments concerning our manuscript entitled “Does financial asset allocation term structure affect audit fees? Evidence from China”. We are happy to respond to your comments and suggestions, and grateful to take this as a great opportunity to update our information and try our best to meet the high standards of PLOS ONE. The main corrections in the paper and the responses to the editor’s and reviewers’ comments, point-by-point, are as follows:

Reviewers' comments:

R: Thank you so much! 

We believe that your valuable comments are very helpful in improving the quality of the manuscript and have expanded our research vision and research ideas, giving us a deeper knowledge of current research. It helps to improve the level of our manuscript.

Reviewer #1: 

1.The appropriateness of the term "Audit fees theory" and whether it is clearly defined in the existing literature needs further investigation. It is suggested that the authors provide a more detailed explanation of this theory and review relevant literature.

R: Thank you so much! 

Thank you for your careful review. The presence of the term audit fee theory in the article is a presentation and translation error. We have revised the expression in the manuscript and adjusted it in the article, and the term "Audit fees theory" was deleted. What we changed in the manuscript is as follows:

Chinese real enterprises face a dilemma characterized by shrinking markets, intensifying competition, overcapacity, profit compression, and declining returns on investment [1, 2]. In the context of risk-oriented auditing, scholars have increasingly explored the relationship between financial asset allocation and audit fees. Audit fees primarily depend on two factors: cost inputs and the risk premium associated with the audit process [3].

2.The significance of conducting the study in the context of China should be further emphasized in the contribution section.

R: Thank you so much! 

Thanks for your comments, we have changed the Contributions section of the Introduction, and we also have changed the expression of theoretical and practical implications sections of the Conclusions. What we changed in the manuscript is as follows:

Introduction:

This study makes several potential contributions. Firstly, it examines the relationship between financial asset allocation and audit fees from two dimensions: short-term and long-term financial assets, thereby extending the existing research on the relationship between financial asset allocation and audit fees in Chinese firms. Secondly, from the perspective of risk-oriented audit fees, this study finds that within the financial asset allocation term structure, long-term financial asset allocation exposes firms to higher financial risks, which is the primary driver of increased audit fees. This finding provides a rational explanation for the differences in audit fee levels among Chinese listed firms from the perspective of risk-oriented, enriching the literature on the factors influencing audit fee variations. Thirdly, it expands the research on independent director characteristics by examining their role in decision-making and monitoring during firms' financial asset allocation. It also offers empirical insights into preventing the risks associated with financial asset allocation in Chinese firms. This study provides a theoretical foundation for understanding the relationship between the financial asset allocation term structure and audit fees in Chinese firms. Additionally, it offers empirical evidence on strategies corporate managers can adopt to mitigate financial asset allocation risks and reduce audit fees. The findings also have practical implications for accounting firms and government agencies.

Conclusions:

China's economy has become an increasingly significant component of global economic development, with Chinese enterprises playing a vital role in driving economic growth. Against the backdrop of the financialization of Chinese real enterprises, this study holds important theoretical and practical implications. First, it extends the discussion on the economic impact of the financial asset allocation term structure of Chinese firms and the determinants of audit costs. Additionally, it uncovers the intrinsic mechanism linking the financial asset allocation term structure to audit fees, emphasizing the need for firms to focus on optimizing their financial asset allocation—particularly long-term financial assets—to mitigate financial risk and reduce audit fees. Furthermore, the study provides a theoretical foundation for strengthening the role of independent directors in Chinese enterprises. Firms should leverage the oversight and decision-making capabilities of independent directors to mitigate risks associated with financial asset allocation, thereby reducing audit supervision and related costs. Accounting firms are encouraged to prioritize understanding the financial asset allocation term structure to determine audit fees more effectively. Lastly, government authorities should guide real enterprises in optimizing their financial asset allocation structures to foster a virtuous cycle between finance and the real economy.

3.The current endogeneity tests in the paper appears inadequate. The authors are encouraged to add one or two more endogeneity tests.

R: Thank you so much!

Thank you for your comment, we have utilized instrumental variable method and GMM for endogeneity test, and the above two methods explain the endogeneity problem of the article. 

Instrumental variable method (IV) refers to the literature Wang H, Cao Y, Yang Q, Yang Z. Does the Financialization of Non-financial Enterprises Promote or Inhibit Corporate Innovation Nankai Business Review. 2017;20(01):155-66. and Dong X, Sun W. Enterprise Financialization,Internal Control and Audit Quality. Journal of Audit & Economics. 2021;36(01):26-36. https://link.cnki.net/urlid/32.1317.F.20210207.1346.024

GMM refers to the literature Zhou S, Ye N, Zhan W. The impact of corporate financialization on corporate innovation:Moderating role of digital transformation. Statistics & Decision. 2024;40(16):183-8.http://dx.doi.org/10.13546/j.cnki.tjyjc.2024.16.033

Meanwhile, according to your comment, we added high-dimensional fixed effects and fixed effects considering individual, industry and year in the endogeneity test. To increase the robustness of the findings, we adjusted the sample interval in the robustness test to reduce the effect of the 2008 financial crisis. And the independent variables are lagged one and two periods for the test. And the manuscript was revised. Detailed endogeneity test and robustness test including Instrumental variable method(IV), GMM, High-dimensional Fixed Effect Model, Replacing the independent variables, Lagging the independent variables by one period and two periods, Fixed effect model and Replacing the sample intervals were used in the manuscript, which took into account the possible scenarios and explained the possible endogeneity and robustness issues, and the results showed that this study is robust. Thank you for your comments, which provide input on the reliability of our study, which we will continue to improve in subsequent studies.

What we changed in the manuscript is as follows:

4.3 Robustness testing

4.3.1 Replacement of measures of the independent variable. Drawing on previous research [27], tthis study excludes loans and advances issued from the calculation of long-term financial assets. The results, presented in Table 4, columns (1)–(2) indicate that the coefficients for SFin1 are insignificant, while the coefficients for CFin1 remain significantly positive at the 1% level. These findings are consistent with the main regression results, reinforcing the robustness of the analysis.

4.3.2 Lagged Independent Variable. Lagged the independent variable by one and two periods, the results are shown in Table 4. Columns (3)-(6) show the results that the coefficients of L1.SFin and L2.SFin are insignificant, the coefficients of L1.CFin and L2.CFin are significantly positive at the 1% level and 5% level. These findings confirm the robustness of the main regression results.

4.3.3 Adjusting the sample interval. To mitigate the potential impact of the financial crisis, data from 2009 and 2010 were excluded from the analysis. The results, presented in columns (7)-(8) of Table 4, show that the coefficients for CFin remain significantly positive at the 1% level. These findings further confirm the robustness of the main regression results.

Table 4

 （1） （2） （3） （4） （5） （6） （7） （8）

Variables LNFEE LNFEE LNFEE LNFEE LNFEE LNFEE LNFEE LNFEE

SFin1 0.250 

 (0.84) 

CFin1 0.458*** 

 (9.06) 

L1.SFin 0.203 

 (1.02) 

L1.CFin 0.319*** 

 (3.47) 

L2.SFin 0.143 

 (0.28) 

L2.CFin 0.204** 

 (2.01) 

SFin 0.172 

 (0.69) 

CFin 0.134***

 (2.99)

Cons 4.088*** 4.064*** 3.875*** 3.927*** 3.106*** 3.437*** 2.517*** 3.313***

 (33.63) (33.61) (28.36) (27.54) (22.61) (24.58) (18.54) (23.16)

Controls yes yes yes yes yes yes yes yes

Ind/Year yes yes yes yes yes yes yes yes

N 8547 8547 8547 8547 8547 8547 7942 7942

R2 0.687 0.690 0.514 0.506 0.462 0.493 0.416 0.484

Note: t-statistics in parentheses, *, **, *** denote statistically significant at 10%, 5%, and 1%.

4.3.4 Endogeneity test. 

Instrumental Variables (IV) Estimation. Following prior research [1, 27], "investment return/net profit" was selected as the instrumental variable (IV) to address potential endogeneity concerns in the study's findings. Investment return is strongly correlated with the level of financial asset allocation but has no intrinsic relationship with auditor input or effort. This ensures that the IV is uncorrelated with audit fees, satisfying the basic conditions for a valid instrumental variable. At the same time, Hansen J statistic P value is 0.167, F value is 549.457, LM statistic is 206.592, confirm the validity of the instrumental variable. Columns (1) and (2) in Table 5 present the results of the IV method. In the first-stage regression, the coefficient of the IV in column (1) is 0.0219, which is significantly positive at the 1% level, indicating a strong positive relationship between the IV and the endogenous variable. In the second-stage regression, the coefficient of long-term financial assets in column (2) is 0.301, also significantly positive at the 1% level, demonstrating that audit pricing increases substantially with higher levels of long-term financial assets. These results are consistent with the main regression findings. 

Generalized Method of Moments(GMM). GMM was used to re-evaluate the results. Following prior research, the lagged one-period independent variable was selected as the instrumental variable [36]. The results satisfy the requirements for overidentification and autocorrelation tests. As shown in Table 5, column (3) reports a regression coefficient of 0.109 for short-term financial assets, which is not statistically significant. Column (4) indicates that the regression coefficient for long-term financial assets is significantly positive at the 5% level, further confirming the reliability of the previous findings

Fixed Effects Model. Endogeneity can arise from omitted variables. To address this, the study incorporates year fixed effects, industry fixed effects, and firm fixed effects in the regression model. The results are presented in Table 5. In Column (5), the coefficient for short-term financial assets is insignificant, while in Column (6), the coefficient is significantly positive at the 5% level. Additionally, a high-dimensional fixed effects regression at the "industry-year" level is conducted, as shown in Columns (7) and (8) of Table 5. The results indicate that higher levels of long-term financial assets significantly increase audit fees, confirming the robustness of the main regression findings.

Table 5

 （1） （2） （3） （4） （5） （6） （7） （8）

 IV-2SLS GMM GMM 

Variables CFin LNFEE LNFEE LNFEE LNFEE LNFEE LNFEE LNFEE

IV 0.0219*** 

 (12.78) 

CFin 0.301*** 

 (5.13) 

SFin 0.109 0.312 0.407 

 (0.96) (1.16) (1.27) 

CFin 0.016** 0.330** 0.493**

 (1.99) (1.97) (2.41)

Cons 0.0571** 4.014*** 1.976** 2.013*** 1.476*** 3.540*** 2.219*** 3.131***

 (2.20) (27.70) (2.03) (16.03) (23.54) (26.18) (18.33) (19.16)

Controls yes yes yes yes yes yes yes yes

Ind/Year yes yes yes yes yes yes yes yes

Company no no no no yes yes no no

Ind -Year no no no no no no yes yes

N 8547 8547 8547 8547 8547 8547 8547 8547

R2 0.246 0.680 0.427 0.549 0.436 0.548 0.477 0.501

Note: t-statistics in parentheses, *, **, *** denote statistically significant at 10%, 5%, and 1%.

4.The title and introduction of the paper place significant emphasis on the influence of independent directors. However, in the mechanism analysis section, the characteristics of independent directors are only considered as one aspect of the analysis. It is suggested that the authors either include the analysis of independent directors' characteristics in the hypothesis development section, or adjust the title and introduction to focus specifically on audit fees and financial asset allocation structures. This would improve the clarity and logical coherence of the paper’s structure.

R: Thank you so much! 

We have restructured the title and introduction of the article, we adjusted the article title to “Does financial asset allocation term structure affect audit fees? Evidence from China”, we condensed the analysis of independent directors characteristics in the introduction, and streamlined the analysis of independent director characteristics in the Mechanisms Analysis section. We improved the clarity and logical coherence of the paper’s structure. 

What we changed in the manuscript is as follows:

Article title:

“Does financial asset allocation term structure affect audit fees? Evidence from China”

The independent directors characteristics are condensed in the introduction.

In recent years, Chinese government and investors have increasingly emphasized the monitoring and decision-making roles of independent directors. The governance of independent directors influences the relationship between the financial asset allocation term structure and audit fees. However, the direction and magnitude of this influence depend on the characteristics of the independent directors. Consequently, different characteristics of independent directors may have varying impacts on the relationship between the financial asset allocation term structure and corporate audit fees [14]. 

Meanwhile, We have streamlined the analysis of independent director characteristics in the Mechanisms Analysis section

4.4.2 Independent directors characteristics

Independent directors play a critical role in strategy formulation and addressing procedural uncertainties [37-39]. Effective governance by independent directors enhances corporate decision-making efficiency, improves asset allocation decisions, and optimizes the overall corporate asset structure [40]. The characteristics of independent directors define the structure and distribution of their authority, ensuring professional and independent decision-making and monitoring within the framework of formal governance while safeguarding the interests of all shareholders [37]. Specifically, their governance mechanism enhances both decision-making and oversight capacities. According to higher-order theory, the characteristics of top managers, such as age, gender, educational background, and career experiences, which significantly influence a firm’s strategic choices [41]. This study aims to investigate how the individual characteristics of independent directors moderate the relationship between long-term financial assets and audit fees by improving decision-making efficiency. This study identifies gender, financial experience, overseas experience, and academic background as key proxies for the individual characteristics of independent directors. Using Model (2), it examines the moderating effects of the proportion of women (FID), the proportion of directors with financial backgrou

---

## [Editor Report · Decision Letter 2]

3 Jan 2025

Does financial asset allocation term structure affect audit fees? Evidence from China

PONE-D-24-38347R2

Dear Dr. Nie,

We’re pleased to inform you that your manuscript has been judged scientifically suitable for publication and will be formally accepted for publication once it meets all outstanding technical requirements.

Kind regards,

Florian Follert

Academic Editor

PLOS ONE

---

## [Editor Report · Acceptance letter]

8 Jan 2025

PONE-D-24-38347R2 

PLOS ONE

Dear Dr. Nie, 

I'm pleased to inform you that your manuscript has been deemed suitable for publication in PLOS ONE. Congratulations! Your manuscript is now being handed over to our production team.

Kind regards, 

on behalf of

Prof. Dr. Florian Follert 

Academic Editor

PLOS ONE